# Mixed-Species *Acacia* Plantation Decreases Soil Organic Carbon and Total Nitrogen Concentrations but Favors Species Regeneration and Tree Growth over Monoculture: A Thirty-Three-Year Field Experiment in Southern China

Shengnan Ouyang [1,2,*], Liehua Tie [1], Xingquan Rao [2], Xi'an Cai [2], Suping Liu [2], Valentina Vitali [3], Lanying Wei [4], Qingshui Yu [5], Dan Sun [2], Yongbiao Lin [2], Arun K. Bose [3,6], Arthur Gessler [3,7] and Weijun Shen [8]

[1] Institute for Forest Resources and Environment of Guizhou, Key Laboratory of Forest Cultivation in Plateau Mountain of Guizhou Province, College of Forestry, Guizhou University, Guiyang 550025, China; lhtie@gzu.edu.cn
[2] Key Laboratory of Vegetation Restoration and Management of Degraded Ecosystem, South China Botanical Garden, Chinese Academy of Sciences, Guangzhou 510650, China
[3] Swiss Federal Institute for Forest, Snow and Landscape Research, 8903 Birmensdorf, Switzerland; valentina.vitali@wsl.ch (V.V.); arun.bose@wsl.ch (A.K.B.); arthur.gessler@wsl.ch (A.G.)
[4] Guangxi Forest Inventory and Planning Institution, Nanning 530011, China; weilanyingccn@163.com
[5] College of Urban and Environmental Sciences, Peking University, Beijing 100871, China; yuqingshui@pku.edu.cn
[6] Forestry and Wood Technology Discipline, Khulna University, Khulna 9208, Bangladesh
[7] Institute of Terrestrial Ecosystems, ETH Zurich, 8902 Zurich, Switzerland
[8] Guangxi Key Laboratory of Forest Ecology and Conservation, State Key Laboratory for Conservation and Utilization of Agro-Bioresources, College of Forestry, Guangxi University, Nanning 530004, China; shenweijun@gxu.edu.cn
* Correspondence: snouyang@gzu.edu.cn; Tel.: +86-13535565034

**Abstract:** Mixed-species plantations of trees with N-fixing species have the potential of promoting forest productivity and soil fertility. However, few studies in the literature have addressed the advantages of mixed-species plantations of leguminous trees over monocultures of leguminous trees based on in situ inventories over a long time period. Here, we monitored the dynamics of tree community composition, vegetation biomass, soil nutrients, and soil microbial phospholipid fatty acids (PLFAs), in an *Acacia mangium* monoculture plantation during 33 years of development and compared it with a mixed-species plantation of *A. mangium* associated with 56 native species which were underplanted 14 years after the initial establishment. Leaf N and phosphorus (P) concentrations of three main species in the overstory and understory of the *A. mangium* monoculture were measured. Our results showed that the soil organic carbon (SOC), total nitrogen (TN), and available phosphorus (AP) concentrations significantly increased over time during the approximately thirty years of *A. mangium* monoculture plantation, while the disadvantages were associated with new species regeneration and the increment of vegetation biomass. In the *A. mangium* monoculture plantation, leaf N concentration of *A. mangium*, *Rhodomyrtus tomentosa*, and *Dicranopteris dichotoma* continuously increased from 21 to 31 years, while the leaf P concentration of *A. mangium* and *R. tomentosa* decreased. The mixed-species plantations of *A. mangium* with native tree species had lower SOC and soil TN concentrations, more new tree species recruitment in the understory, and faster vegetation biomass increment than the *A. mangium* monoculture. However, the PLFAs of soil microbial groups were slightly different between the two types of plantations. We conclude that improved soil N nutrient condition by *A. mangium* monoculture benefits N absorption by *A. mangium*, *R. tomentosa*, and *D. tomentosa*, while low soil AP limits P absorption by *A. mangium* and *R. tomentosa*. Meanwhile, transforming the *A. mangium* monoculture into a mixed-species plantation via the introduction of multiple native species into the *A. mangium* monoculture decreases SOC and TN concentrations but the advantages include improving forest regeneration and maintaining forest growth in a long-term sequence. These findings provide useful and practical suggestions for managing forest monocultures of *A. mangium* in subtropical regions.

**Keywords:** *Acacia mangium* monoculture; mixed-species plantation; species regeneration; soil microbial community; soil organic carbon

## 1. Introduction

The area of planted forests across the globe covers 294 million hectares and is still increasing to balance the loss of natural forests [1–3]. The majority of these planted forests are homogenous in composition with only one fast-growing tree species to optimize wood and fiber production [4,5]. It has been found that almost half of the subtropical forest area is currently under monoculture (single tree species plantation) [6]. However, several recent studies have identified problems with monocultures, for example, exacerbating the loss of biodiversity, decreasing productivity, increasing the risk of pest and disease infection, reducing soil fertility, and increasing vulnerability to climate change [4,7,8]. These constraints could result in a loss of economic and ecological functions across forests [9,10].

Studies have proposed that mixed-species plantations have greater levels of ecosystem functions and services, including higher productivity and soil organic matter concentration, than monocultures [11,12]. Especially, mixed-species plantations of trees with nitrogen-fixing species have been recommended, with the additional advantages that they can reduce the need for nitrogen (N) fertilization [13,14], decrease competition for light, and accelerate the soil N mineralization and phosphorus (P) cycle [14–16]. However, evidence from numerous studies has proven that the production of mixed-species plantations is lower than that in monocultures due to erroneous selection of trees species for the mixtures and unsuitable sites for planting [15,17,18]. Moreover, in recent decades, mixed-species plantations with native trees have been recommended over those with exotic trees [14,19,20]. For example, from 1990 to 2015, more than 80% of the mixed-species plantation area was comprised of native trees [2]. An increasing number of studies have shown that mixed-species plantations with native trees have higher productivity and nutrient use efficiency relative to monocultures [18,21]. Nonetheless, previous studies of mixed-species plantations have mostly focused on mixtures of from two to three trees [14,17]. Until now, few experiments had studied the differences between an exotic N-fixing species monoculture and a mixed-species plantation with dozens of native trees, in terms of plant species composition; soil organic carbon (SOC), nutrient (e.g., nitrate ($NO_3^-$-N), ammonium ($NH_4^+$-N), and soil total phosphorus) concentrations; and soil microbial community, and especially, the long-term effects remain unclear.

In southern China (tropical and subtropical areas), the forests have historically been damaged by timber harvesting, improper logging, and firewood collection [22,23]. To promote forest recovery and soil amelioration, an initiative, namely monoculture planting of *Acacia mangium* (a N-fixing tree species, native to Australia and Papua New Guinea), was implemented in the 1980s [24,25]. However, recent studies have documented that the *A. mangium* monocultures in southern China had poor understory regeneration [26,27], low biodiversity, and lacked stableness [25]. Therefore, transforming the *A. mangium* monocultures to mixed-species plantations with dozens of native trees was initiated and applied within forest management activities [24,28]. However, the long-term effects of *A. mangium* monocultures in southern China, especially the differences between *A. mangium* monocultures and mixed-species plantations with native trees, are still unclear.

Hence, in the current study, we aim (1) to evaluate the effect of the planting year on the tree community composition, vegetation biomass, soil nutrient properties, and foliar N and P concentrations of the trees, shrubs, and herbs in an *A. mangium* monoculture using long-term inventory data and (2) to clarify the effect of the plantation type (*A. mangium* monoculture and *A. mangium* mixed-species plantation with native trees) on tree community composition, vegetation biomass, soil physiochemical properties, and soil microbial community structure. Based on the above-mentioned goals, we attempt to test the following hypotheses: (1) The plant species richness, foliar N and P concentrations,

and the concentration of SOC and other soil nutrients in the *A. mangium* monoculture will increase with the increment of planting year. This is based on the fact that previous studies have shown that leguminous N-fixing trees can enhance SOC stock, soil N availability, and the rate of soil P cycle [14,16,29,30]; (2) mixed-species plantation with native trees will favor new species recruitment and increase vegetation biomass, soil nutrient contents, and soil microbial PLFAs compared to *A. mangium* monoculture. This is based on the fact that previous studies have shown that mixed-species plantations have higher productivity and nutrient use efficiency relative to monocultures [18,29,31]. The in-situ inventories of the stands over a long-time sequence, in this study, can improve our understanding of the long-term effects of N-fixing tree monoculture and N-fixing mixed-species plantation with dozens of native trees on the plant soil and soil microbial community of the stands.

## 2. Materials and Methods

### 2.1. Study Site

The study was conducted at the Heshan National Field Research Station (60.7 m a.s.l., 22°34′ N, 112°50′ E, Figure 1A) in the subtropical region of southern China. The climate in this region is characterized by a subtropical monsoon associated with a wet season from April to September during which 80% of the 1700 mm of annual precipitation falls. In the past 30 years, the mean annual temperature has been 21.7 °C. The soil is classified as acrisols based on WRB soil classification [32], which is a type of heavily acidic and easily leached soil. The zonal climax vegetation is the subtropical evergreen broad-leaf forest. As a result of long-term anthrophonic disturbances, such as logging, burning, and harvesting of wood for biofuel production, the soil in this area has eroded and the original vegetation has almost disappeared, leading to vast areas of degraded land [22]. The site was covered by grass species before the establishment of the experiment.

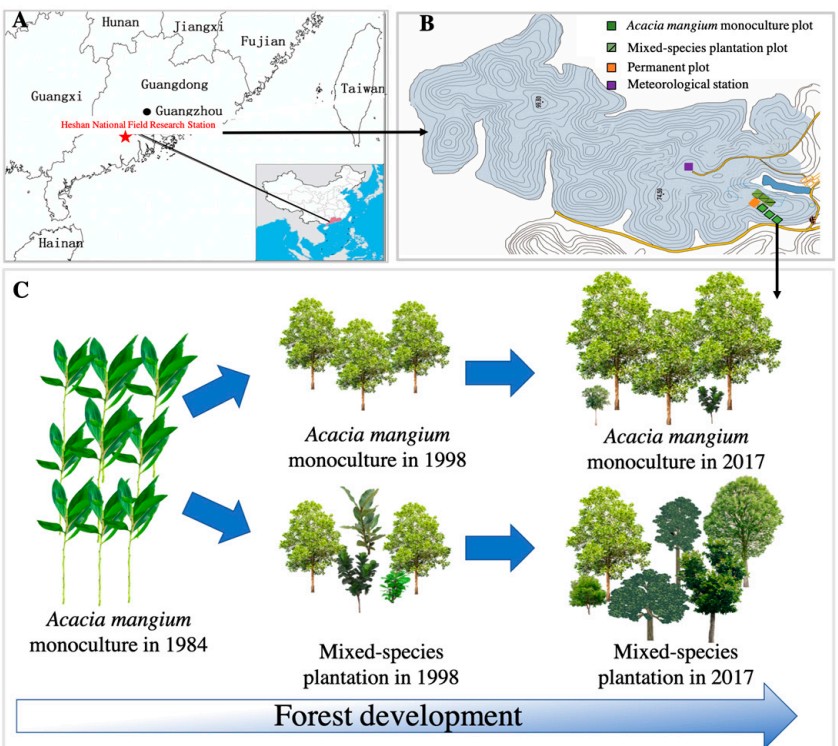

**Figure 1.** Schematic graph of the experimental setup: (**A**) The location of the site; (**B**) the distribution setup and distribution of the plot within the study site; (**C**) the time point of planting the *A. mangium* monoculture in 1984, the introduction of native species into the *A. mangium* monoculture in 1998, and the inventory of the *A. mangium* monoculture and the *A. mangium* mixed-species plantation in 2017.

### 2.2. Experimental Design

2.2.1. Plot Design

In 1984, a pure plantation of *Acacia mangium* (an exotic leguminous species) was established on an area of about 5 hectares of homogenously degraded hilly land to rehabilitate the degraded ecosystem (Figure 1B,C). Seedlings of *A. mangium* were planted at a spacing of 2.5 m × 3 m, with a basal fertilizer application of 50 g of potassium chloride and 50 g of urea [22]. In 1998, an area of 2.5 hectares of the *A. mangium* monoculture was selected, then, approximately 50% of the *A. mangium* trees were randomly cut down, and subsequently, one- or two-year-old seedlings of 56 native species were randomly planted in the interspaces (see Table S1 for the list of the native species).

A permanent quadrat (hereafter referred to as a permanent plot) with an area of 400 m$^2$ (20 × 20 m) in the *A. mangium* monoculture was set up simultaneously in 1984 when the experiment was set up (Figure 1B). The objective of establishing the permanent plot was to monitor the dynamics of tree community structure and soil nutrients over time in the *A. mangium* monoculture. Another six separate plots were established in 1998. Each plot had an area of 400 m$^2$ (20 × 20 m) and the plots were separated from each other by at least 50 m. Three of the separated plots were established as *A. mangium* monoculture (hereafter referred to as the monoculture plots) and the other three plots were established as mixed-species plantations of *A. mangium* with native species (hereafter referred to as the mixed-species plantation plots). The objectives of establishing six plots were to study the differences in tree community structure, soil physiochemical properties, and soil microbial community structure between *A. mangium* monoculture andmixed-species plantation of *A. mangium*. Meanwhile, a meteorological station located just outside of the plantation area was established to record daily rainfall and temperature from 1985 to 2015.

2.2.2. Soil and Leaf Sample Collections

Soil and leaf samples that had been collected from the permanent plot of *A. mangium* monoculture over 31 years were used to estimate the roles of *A. mangium* monoculture in recovering the degenerated area. To avoid disturbing soils of the permanent plot during the inventory, soil samples were collected at a location close to the permanent plot. The possible differences in soil properties between the permanent plot and the location close to the permanent plot should have been neglectable, as the *A. mangium* monoculture was established at the same time (1984).

Soil sampling in the permanent plot was conducted in 1986, 1990, 1994, 2002, 2005, 2010, and 2015 to study the dynamics of soil nutrient properties (i.e., SOC, total nitrogen (TN), and available phosphorus (AP) and the ratios of SOC to TN, SOC to total phosphorus (TP), and TN to TP) in the *A. mangium* monoculture during 31 years of planting (Table S2). All soil samples were collected from 0 to 20 cm depths using an auger ($\Phi$ = 35 mm). Four independently collected soil cores from the same topography were mixed to form one composite sample, and four composite samples were collected each time. All soil samples were sieved through 2 mm mesh to remove roots, litter, and stones. *Rhodomyrtus tomentosa* is a shrub and *Dicranopteris dichotoma* is a fern; both *R. tomentosa* and *D. dichotoma* grew in the understory of the *A. mangium* monoculture. Leaf samples of *A. mangium* as well as leaf samples of *R. tomentosa* and *D. dichotoma* were collected from individuals in the permanent plot which corresponded to the soil sampling in 2005, 2010, and 2015. Fresh, well-developed, and fully expanded leaves of each species were harvested. Leaf samples were collected from 3 to 5 individuals to form one composite sample, and four composite samples were collected each time. All leaf samples were dried to a constant mass at 65 °C, and then, ground to powder for the leaf N and P contents analysis.

Soil samples from the three monoculture plots and three mixed-species plots were collected in August 2011 and 2017 to study the differences in soil microbial community structure and soil physiochemical properties (i.e., dissolved organic carbon (DOC), TN, $NO_3^-$-N, $NH_4^+$-N, TP, and pH) between *A. mangium* monoculture and *A. mangium* mixed-species plantation with native trees. Soil samples were collected in the same way as those

in the permanent plot of *A. mangium* monoculture. At each sampling time, the soil samples collected from the six plots were divided into three subsamples for further analysis: The first subsample was air-dried and stored at room temperature for the measurement of soil physiochemical properties (SOC, DOC, TN, TP, pH); the second subsample was stored at 4 °C for soil inorganic N ($NH_4^+$-N and $NO_3^-$-N) analyses; the third subsample was stored at −20 °C after freezing-drying for phospholipid fatty acids (PLFAs) analysis.

### 2.2.3. Soil and Leaf Sample Property Analyses

The concentration of SOC was determined by following the potassium dichromate oxidation method described by [33]. Soil TN concentration was assayed by semi-micro Kjeldahl digestion [34]. The soil AP concentration was determined by following the "Bray-1" available P method [35]. Soil AP was extracted with a solution of 0.03 M ammonium fluoride and 0.025 M hydrochloric acid and measured using the colorimetric method [34]. Soil TP concentration was determined by the concentrated sulfuric acid and perchloric acid digestion method [36]. Fresh soil samples were filtered with 2 M KCl, and all of the filtered extracts were analyzed for $NH_4^+$-N and $NO_3^-$-N with a flow injection autoanalyzer (FIA, Lachat Instruments, Loveland, CO, USA). The soil DOC concentration was extracted with 0.5 M $K_2SO_4$, and the extracts were analyzed using a TOC analyzer (TOC-5000, Shimadzu, Kyoto, Japan). Soil pH was measured in a 1:2.5 mixture suspension of soil/water using a pH meter (Horiba F-71S, Horiba, Kyoto, Japan). The soil C to N ratio (C:N), C to P ratio (C:P), and N to P ratio (N:P) were calculated with the ratios of SOC to TN concentration, SOC to TP concentration, and TN to TP concentration, respectively.

Phospholipid fatty acids were extracted from the soil samples according to method by Frostegård et al. [37], and the method of analyzing microbial PLFAs was modified from Frostegård and Bååth [38]. Briefly, total lipids were extracted overnight from 8 g of freeze-dried soil in a chloroform/methanol/phosphate buffer mixture (1:2:0.8, *v/v/v*), and then, the extracted lipids were transferred to a solid-phase silica column (Agilent Technologies, Palo Alto, CA, USA). The phospholipids were separated from neutral lipids, glycolipids, and polar lipids, by alternately using 5 mL of chloroform and 10 mL of acetone or 5 mL of methanol in a silica column. After separation, the polar lipid fraction was transferred into fatty acid methyl eaters by using 1 mL 0.2 M methylated KOH. The fatty acid methyl esters were dissolved in an internal standard-methyl nonadecanoate fatty acid (19:0), and then, they were identified and measured using an Agilent 6890 gas chromatograph mass spectrometer (Agilent Technologies, Palo Alto, CA, USA).

According to the fatty acid nomenclature described by Frostegård and Bååth [38], the sum of a14:0, 14:0, i15:0, a15:0, 15:0, i16:0, 16:0, i17:0, cy17:0, 17:0, 16 1$\omega$7c, 18:1$\omega$7c, 18:0, and cy19:0$\omega$8c was considered to be an indicator of the bacterial group [39,40]. The sum of saturated unsubstituted fatty acids i14:0, i15:0, a15:0, i16:0, i17:0, and a17:0 represented Gram-positive bacteria [41], while the sum of monounsaturated fatty acid, 16:1$\omega$7c, cy17:0, 18:1$\omega$7c, and cy19:0$\omega$8c indicated Gram-negative bacteria [42,43]. Four fatty acids (16:1$\omega$5c, 18:1$\omega$9c, 18:2$\omega$6, 9c, and 18:3w6,9c) were chosen to represent the fungi group, and one fatty acid (i.e., 16:1$\omega$5c) was considered to be arbuscular mycorrhizal fungi (AMF) [44]. The PLFAs 10Me 16:0, 10Me 17:0, and 10Me 18:0 were considered to be representative of actinomycetes [45]. In addition, all of the PLFAs mentioned above and other PLFAs were summed as the total PLFAs of the soil microbial community. All the PLFAs were calculated as nanomoles per gram of dry soil as the absolute abundance of lipids. The relative abundance of individual lipids was expressed in relative mol%, calculated with moles of a particular PLFA divided by the total amount of PLFAs (moles) in a sample. The sum of the individual relative abundance represented the relative abundance of particular microbial groups (e.g., bacterial, Gram-positive bacteria, Gram-negative bacteria, fungi, and actinomycetes).

The foliar total nitrogen (N) and total phosphorus (P) concentrations of *A. mangium*, *R. tomentosa,* and *D. dichotoma* were analyzed. The foliar total N concentration was determined by using a CHNS/O analyzer (PerkinElmer II2400, Perkin Elmer Inc., Waltham,

MA, USA). The foliar total P concentration was determined by following the concentrated sulfuric acid and perchloric acid digestion method [46]. Different tree species biomass was calculated by allometric equations (Table S3).

### 2.3. Statistical Analyses

All data passed the test of normality. A one-way ANOVA analysis with Tukey's HSD test was performed to determine the effects of the *A. mangium* planting year on the concentrations of SOC, soil TN, and soil AP; the ratios of soil C:N, C:P, and N:P; and the concentrations of foliar TN and foliar TP in the permanent plot. Linear regressions were performed to test the relationships of SOC concentration, soil TN concentration, soil C:P ratio, and soil N:P ratio with the planting year in the permanent plot of *A. mangium* monoculture. A two-way ANOVA analysis with Tukey's HSD test was used to determine the effects of the planting year, plantation type (monoculture and mixed-species plantation), and their interactions on soil physiochemical properties and microbial community structure (based on the PLFA results of different microbial groups) in the monoculture plots and mixed-species plantation plots. A redundancy analysis (RDA) was used to test the relationships between soil physiochemical properties and the relative abundances of soil microbial groups. The RDA was performed using the CANOCO software for Windows 5.0 (Ithaca, NY, USA). SPSS 19 (SPSS, Inc., Chicago, IL, USA) was used for all statistical analyses. The statistical significance was determined at the 0.05 level.

## 3. Results

### 3.1. Tree Community Composition, Vegetation Biomass, and Soil and Foliar Properties in the A. mangium Monoculture

The number of regenerated tree species in the *A. mangium* monoculture decreased in 2000 compared to the number recorded in 1995 and in 1997, showing the lowest value during the 31 years of observation (Figure 2B). However, the number of regenerated tree species increased in 2012 compared with that in 2003. The understory shrub and herb species richness were less than seven after 31 years of *A. mangium* planting (Table S1).

The mean DBH of the *A. mangium* trees generally increased with the increment of the planting year (Figure 2C). In addition, there was a decrease in the density and biomass of *A. mangium* trees from 2000 to 2003 and an increase in both parameters from 2003 to 2007.

The soil organic carbon and TN concentrations significantly increased over the 31 years of observation (Figure 3A,B); the SOC concentration increased by 2.5-fold compared with the beginning of *A. mangium* planting and the soil TN concentration increased by 3-fold. Moreover, the linear regression analysis showed that the SOC ($r^2$ = 0.91, $p$ = 0.000) and soil TN ($r^2$ = 0.91 and p = 0.000) concentrations were positively related to the increment of *A. mangium* planting year. The soil AP concentration significantly increased from 0.32 mg kg$^{-1}$ (in 1986) to 2.53 mg kg$^{-1}$ (in 2015) ($p$ = 0.003, Figure 3C). In addition, the linear regression analysis showed that soil C:P ratios were positively related to the *A. mangium* planting year ($r^2$ = 0.72, $p$ = 0.008, Figure 3E). The ratio of soil N:P showed a slightly increased trend with the increment of *A. mangium* planting year ($r^2$ = 0.48, $p$ = 0.056, Figure 3F).

The foliar N concentrations of *A. mangium*, *R. tomentosa*, and *D. dichotoma* significantly increased from 2005 to 2010 (i.e., from 21 to 31 years after planting *A. mangium*) (Figure 4A). The foliar P concentrations of *A. mangium* and *R. tomentosa* were significantly lower in 2010 and in 2015 than those in 2005, while that of *D. dichotoma* showed a contrary trend (Figure 4B).

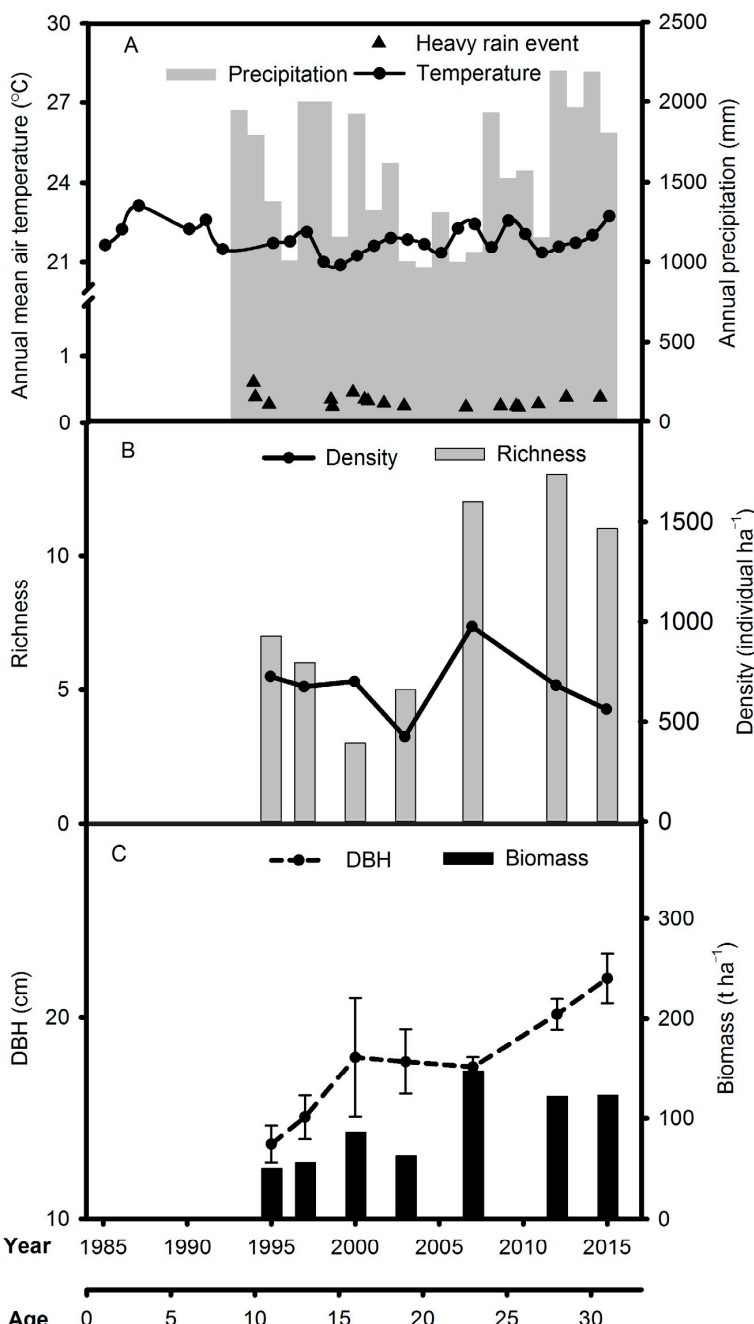

**Figure 2.** Meteorological records and forest inventory after the initiation of the *A. mangium* plantation: (**A**) The annual mean air temperature, annual precipitation, and heavy rainfall events (daily precipitation > 100 mm) from 1984 to 2015 recorded at the Heshan station in southeastern China; (**B**,**C**) the dynamics of tree density, species richness, DBH (diameter at breast height), and tree biomass in the permanent plot of *A. mangium* monoculture after 31 years of planting (1984 to 2015). Error bars represent the standard deviation of tree species' DBH in the *A. mangium* monoculture.

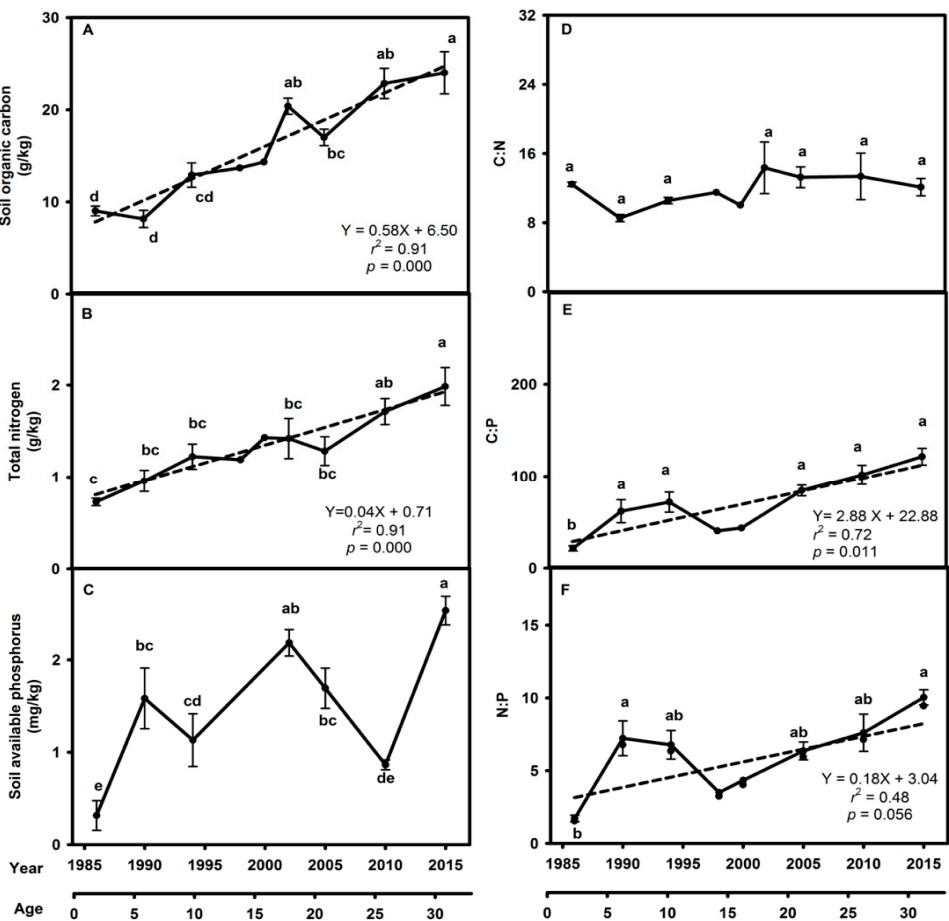

**Figure 3. (A–C)** Dynamics of soil organic carbon, total nitrogen, and available phosphorus concentrations; (**D–F**) the ratio of soil organic carbon to total nitrogen (C:N), soil organic carbon to total phosphorus (C:P), and soil total nitrogen to total phosphorus (N:P), in the *A. mangium* monoculture during 31 years of planting. The black dashed lines in the panel indicate the linear regression with plantation age. Error bars represent the standard error (*n* = 4). Data on soil organic carbon and total nitrogen concentrations in 1998 and 2000 was referred to previous study results [36,46]. Differernt lowercase letters in the figure indicate significant differences between the planting year.

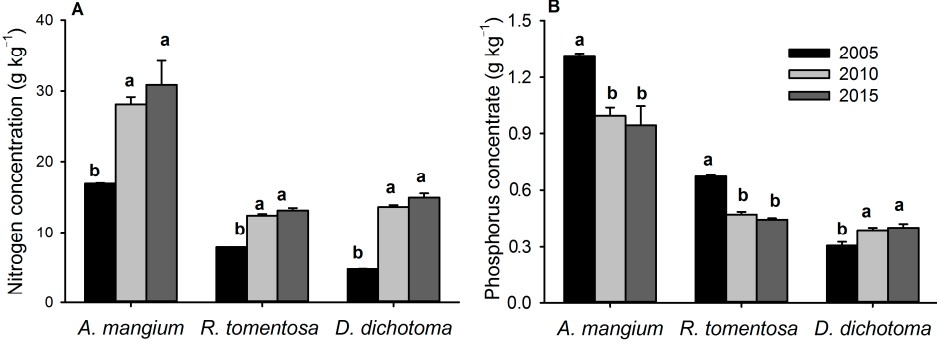

**Figure 4.** Total nitrogen (A) and phosphorus (B) concentrations of *A. mangium*, *R. tomentosa*, and *D. dichotoma* leaves in 2005, 2010, and 2015. The different lowercase letters represent significant differences among the three time points of sampling at *p* < 0.05. Error bars represent the standard error (*n* = 4).

### 3.2. Comparison between the A. mangium Monoculture and the A. mangium Mixed-Species Plantation

A total of 30 among the 56 planted native species together with 8 naturally regenerated species survived in the mixed-species plantation plots in 2017 (Table 1). The understory shrub and herb species richness was higher in the mixed-species plantation plots than in the monoculture plots. The mean DBH of *A. mangium* trees increased by 55.6% from 2011 to 2017 in the mixed-species plantation plots, and the increase was higher than that in the monoculture plots (23.6%). Moreover, the mean DBH and height of the introduced native species and newly recruited species also increased (except the mean height of newly recruited species) in the mixed-species plantation plots, but not in the monoculture plots. The total vegetation biomass in the mixed-species plantation plots increased by 106% from 2011 to 2017, while the total vegetation biomass in the monoculture plots slightly declined during this period. In 2017, the biomass of *A. mangium* trees and the biomass of newly regenerated species and introduced native species in the mixed-species plantation plots contributed to 44.6% and 55.4% of the total vegetation biomass, respectively. In contrast, in the monoculture plots, the biomass of newly regenerated species was negligible compared to the biomass of *A. mangium* trees in 2017.

**Table 1.** The stand density, biomass, species richness, mean DBH, mean height, and shrub and herbaceous species richness in *A. mangium* monoculture and mixed-species plantation of *A. mangium*, in 2011 and in 2017.

| Variables | *A. mangium* Monoculture Plots | | | | | | Mixed-Species Plantation of *A. mangium* Plots | | | | | | | |
| | Total | | *A. mangium* | | Regenerated Species | | Total | | *A. mangium* | | Planted Native Species | | Regenerated Species | |
| | 2011 | 2017 | 2011 | 2017 | 2011 | 2017 | 2011 | 2017 | 2011 | 2017 | 2011 | 2017 | 2011 | 2017 |
|---|---|---|---|---|---|---|---|---|---|---|---|---|---|---|
| Density (individual ha$^{-1}$) | 833 | 634 | 700 | 467 | 133 | 167 | 1287 | 1233 | 125 | 108 | 775 | 900 | 387 | 225 |
| Biomass (t ha$^{-1}$) | 118.87 | 108.69 | 118.76 | 108.58 | 0.10 | 0.10 | 50.06 | 103.17 | 24.73 | 46.04 | 9.06 | 38.29 | 16.81 | 18.77 |
| Tree species richness | 6 | 7 | 1 | 1 | 5 | 6 | 39 | 38 | 1 | 1 | 30 | 30 | 8 | 7 |
| Mean tree DBH (cm) | 14.87 ± 0.89 | 17.54 ± 1.44 | 17.14 ± 0.86 | 21.19 ± 1.39 | 2.98 ± 0.40 | 2.93 ± 0.50 | 7.83 ± 0.64 | 10.00 ± 0.82 | 20.21 ± 2.25 | 31.44 ± 3.35 | 6.66 ± 0.67 | 9.31 ± 0.38 | 2.17 ± 0.24 | 2.45 ± 0.32 |
| Mean tree height (m) | 14.23 ± 0.65 | 12.67 ± 0.79 | 16.39 ± 0.49 | 15.14 ± 0.65 | 2.86 ± 0.30 | 2.81 ± 0.36 | 7.67 ± 0.53 | 8.52 ± 0.46 | 15.40 ± 1.59 | 17.00 ± 0.74 | 7.28 ± 0.65 | 8.67 ± 0.27 | 3.64 ± 0.16 | 3.47 ± 0.33 |
| Shrub and herb richness | 20 | 13 | | | | | 28 | 21 | | | | | | |

In 2011, the concentrations of SOC, soil DOC, and soil TN; the soil pH; and the soil C:P ratio were significantly lower in the mixed-species plantation plots than those in the monoculture plots (Figure 5A–C,F,G), while the soil TP concentration was significantly higher in the mixed-species plantation plots than that in the monoculture plots. In 2017, the concentrations of soil DOC, $NO_3^-$-N, $NH_4^+$-N, and TP were comparable between the mixed-species plantation plots and the monoculture plots (Figure 5B,D–F). In addition, the interaction of the planting year (2011 and 2017) and plantation type (monoculture and mixed-species plantation) had significant effects on soil TN and $NH_4^+$-N concentrations and the soil C:N ratio (Table S4).

The relative abundance of soil microbial PLFAs was insignificantly different between the monoculture plots and the mixed-species plantation plots (Figure 5H–N and Table S5), but was significantly increased by planting year (from 2011 to 2017) in both the monoculture plots and mixed-species plantation plots. In addition, the RDA results showed that soil physiochemical properties (i.e., pH, $NO_3^-$-N:$NH_4^+$-N ratio, and the concentrations of TN, DOC, TP, and SOC) could explain 60.4% of the total variation in the relative abundance of soil microbial PLFAs, including 50.6% at axis 1 and 9.8% at axis 2 (Figure 6). Among them, the soil pH, soil $NO_3^-$-N:$NH_4^+$-N ratio, and soil TN and DOC concentrations significantly contributed to the total variance ($p < 0.016$), while the concentrations of soil TP and SOC showed insignificant contributions ($p > 0.116$). Furthermore, there were negative correlations among the soil pH and the Gram-negative bacteria, AMF, and actinomycetes and among the soil DOC concentration and the Gram-negative bacteria, AMF, and actinomycetes. However, there were positive correlations among the soil TN concentration and

AMF, fungi, and Gram-negative bacteria and among the ratio of soil $NO_3^--N:NH_4^+-N$ and AMF, fungi, and Gram-negative bacteria.

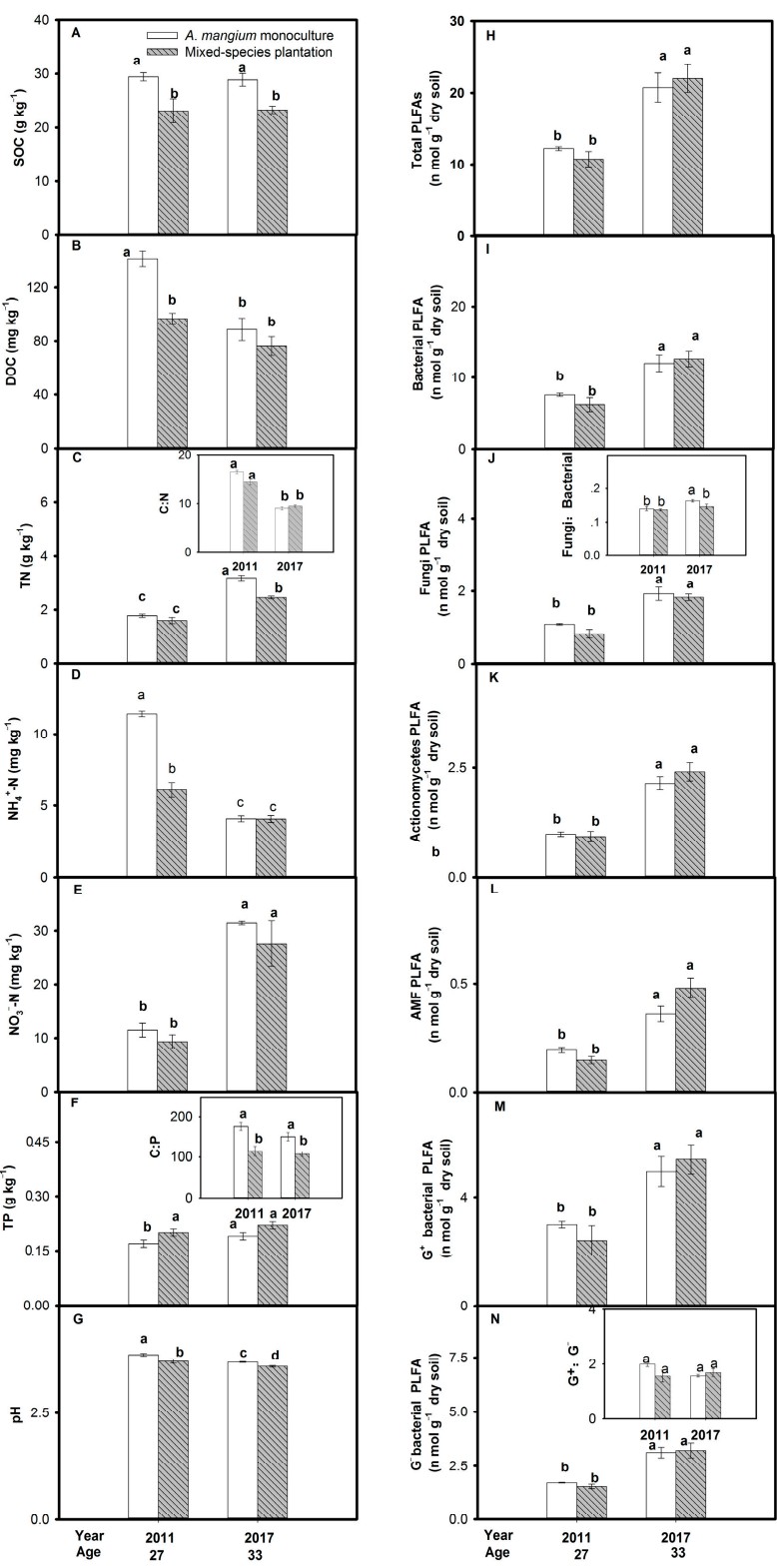

**Figure 5.** Soil physiochemical properties and soil microbial phospholipid fatty acids (PLFAs) in the *A. mangium* monoculture and the *A. mangium* mixed-species plantation in 2011 and in 2017. SOC, DOC,

TN, and TP represent soil organic carbon, dissolved organic carbon, total nitrogen, and total phosphorus, respectively. AMF, G$^+$, and G$^-$ represent arbuscular mycorrhizal fungi, Gram-positive bacteria, and Gram-negative bacteria, respectively. The C:N and C:P represent the ratio of soil organic carbon to total nitrogen and soil organic carbon to total phosphorus, respectively. (**A**–**G**) represent soil organic carbon, dissolved organic carbon, total nitrogen, NH$_4^+$, NO$_3^-$, total phosphorus and pH in the *A. mangium* monoculture and the *A. mangium* mixed-species plantation in 2011 and in 2017, respectively. (**H**–**N**) represent total PLFAs, bacterial PLFA, fungi PLFA, actionmycetes PLFA, arbuscular mycorrhizal fungi PLFA, gram-positive bacteria PLFA and gram-negative bacteria PLFA in the *A. mangium* monoculture and the *A. mangium* mixed-species plantation in 2011 and in 2017, respectively. The micro-panel in (**C**,**F**,**J**,**N**) represent the ratio of soil organic carbon to total nitrogen, soil organic carbon to total phosphorus, fungi PLFA to bacterial PLFA, and gram-positive bacteria PLFA to gram-negative bacteria PLFA in the *A. mangium* monoculture and the *A. mangium* mixed-species plantation in 2011 and in 2017, respectively. Different lowercase letters represent significant differences between the two plantation types at $p < 0.05$ ($n = 3$).

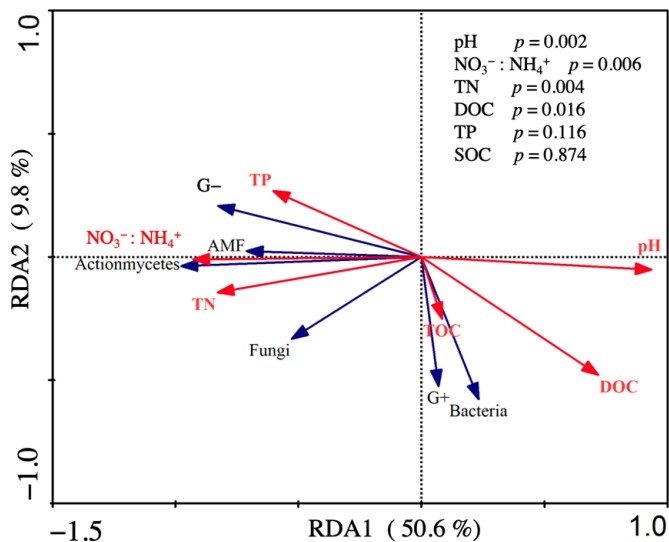

**Figure 6.** Redundancy analysis biplots of phosphorus fatty acid (relative mol%) and soil physiochemical properties in the *A. mangium* monoculture and the *A. mangium* mixed-species plantation. TP, soil total phosphorus concentration; DOC, soil dissolved organic carbon concentration; SOC, soil organic carbon concentration; TN, soil total nitrogen concentration; AMF, soil arbuscular mycorrhizal fungi; G$^+$, soil Gram-positive bacteria; G$^-$, soil Gram-negative bacteria.

## 4. Discussion

### 4.1. Dynamics of the Tree Community Composition, Vegetation Biomass, and Soil and Leaf Properties in the A. mangium Monoculture

In the current study, the mean diameter and biomass of the *A. mangium* trees continuously increased from 1995 to 2000, while unexpected decreases in the *A. mangium* tree density and biomass in the 19th year of planting (i.e., 2003) were recorded (Figure 2B,C). A simultaneous declines in the tree density and biomass of *A. mangium* were most likely related to trees falling, breaking, and even dying due to typhoons and rainstorm events [47]. The meteorological bureau reported that an exceptionally strong typhoon, namely "Dujuan" (with national number 0313 in 2003), occurred on 2 September 2003 with severe damage to the *A. mangium* plantations, and as a consequence, tree density and biomass were decreased. Interestingly, in the following four years (2003–2007) after the unexpected reduction in 2003, the *A. mangium* tree density increased by 1.3-fold (1433 individuals per hectare, Figure 2B), and the species richness also increased (Table S1). On the one hand, the released growth space may have facilitated the germination of *A. mangium* seeds and growth of its seedlings, resulting in an increase in *A. mangium* tree density. On the other

hand, canopy gaps formed as a consequence of trees falling, breaking, and dying would have promoted the recruitment of new species (e.g., eleven species of naturally regenerated trees have been found in 2003, Table S1), thus, increasing the species richness.

In this study, the concentrations of SOC and soil TN significantly increased during the 31 years after planting *A. mangium* (Figure 3A,B). Thus, our first hypothesis about the changes in SOC and other soil nutrients under *A. mangium* monoculture can be accepted. The results are consistent with previous studies on the roles of *A. mangium* in recovering degraded land [16,48,49]. This phenomenon may be related to the acceleration of nutrient cycling [16] and the increase in vegetation biomass [48,49]. As a N-fixing species, *A. mangium* roots can fix atmospheric N with the support of the symbiotic *Rhizobium* [50]. For example, one study pointed out that a rate of 86 kg N ha$^{-1}$ year$^{-1}$ was fixed in an *A. mearnsii* monoculture, resulting in an increase in soil TN concentration [51]. Not only N fixation by roots, but also C- and N-containing compounds in root exudates and litter biomass can contribute to the increment of SOC and TN concentrations [52,53]. Moreover, the foliar N concentrations of *R. tomentosa* and *D. tomentosa* increased in the period from 2005 to 2015 (i.e., from 21 to 31 years after *A. mangium* planting, Figure 4A), indicating that improved soil TN concentration facilitated N uptake by understory non-leguminous shrubs and herbs. The foliar N concentrations of non-leguminous plants increased by the leguminous trees were also found in other studies [14,31,49], which, in turn, might alleviate N competition between non-leguminous plants and leguminous trees [13]. Thus, *A. mangium* monoculture improves soil N nutrient condition, which in return promotes N absorption by *A. mangium* and other species (i.e., *R. tomentosa* and *D. tomentosa*) in the understory.

In this study, the soil AP concentration increased after 31 years of *A. mangium* planting (Figure 3C), implying that planting N-fixing species positively influenced soil P availability. Unexpectedly, the foliar P concentrations of *A. mangium* and *R. tomentosa* decreased with the increment of stand age from 21 to 31 years (Figure 4B), findings which were consistent with studies in subtropical forests [54,55]. On the one hand, subtropical forests are generally limited by P [54]. According to the stoichiometric theory [56], a decrease in foliar P concentration strongly suggested an exacerbating P limitation in this study of subtropical forest. On the other hand, to cope with P limitation, many plants develop special mechanisms to promote soil P availability (e.g., increased acid phosphatase produced by plant roots and soil microorganisms) [57]. Although our results provide limited information from the production of phosphatase enzymes, the relative abundance of AMF was significantly increased with the increment of stand age of *A. mangium* monoculture (Figure 5L). It has been documented that organic acids secreted by arbuscular mycorrhizal fungi (AMF) enhance P availability [58]. Thus, the increased soil AP concentration might relate to the increase in AMF biomass. Overall, the response and adaptation of plants to P limitation can explain the opposite trend between soil AP and foliar P concentrations with the increment of stand age of *A. mangium* monoculture.

### 4.2. Effects of Mixed-Species Plantation on Forest Stand Development

4.2.1. Changes in Tree Community Composition and Vegetation Biomass

In 2017, eight new trees were successfully recruited under the canopy of the mixed-species plantation of *A. mangium* with native trees (Table 1). For example, shade-tolerant and mildly acidic soil-adapted species, such as *Litsea cubeba*, *Melicope pteleifolia*, *Litsea rotundifolia*, and fruit tree species (e.g., *Dimocarpus longan* and *Prunus salicina*) have been found (Table S1). Meanwhile, compared to the *A. mangium* monoculture, the number of new tree regenerations in the mixed-species plantation was higher in 2011 and in 2017 (Table 1). These results support part of our second hypothesis focusing on tree regeneration under mixed-species plantation, suggesting that the introduction of native trees into *A. mangium* plantations is a practical approach for facilitating species regeneration [59,60], because the broadly diversified canopies and heterogeneous habitats in mixed-species plantations can recruit more seed dispersers (e.g., perching and roosting birds), which most likely promote

the greater level of seed dissemination and seedling growth of native trees and understory species [61,62].

In the current study, the vegetation biomass was lower in the mixed-species plantation of *A. mangium* with native trees than that in the *A. mangium* monoculture in 2011 and in 2017 (Table 1), which was contrary to the second hypothesis about the comparison of vegetation biomass between monoculture and mixed-species plantation. This result could be explained by an artificial decrease in the biomass of the trees, since we cut down half of the *A. mangium* trees before establishing the mixed-species plantation. However, the vegetation biomass in the mixed-species plantation largely increased (by 106%) from 2011 to 2017 because of the overall increment of *A. mangium* trees biomass (increased 21.33 t ha$^{-1}$), native tree biomass (increased 29.23 t ha$^{-1}$), and new regenerated plant biomass (increased 1.96 t ha$^{-1}$), although the vegetation biomass in the monoculture slightly declined during this period (Table 1). This result was similar to that of a previous study which reported that mixed-species plantations were generally more productive than the mean of the monocultures [63]. Altogether, the introduction of native species into *A. mangium* plantation can be a promising approach to increase vegetation biomass accumulation.

### 4.2.2. Changes in Soil Physiochemical Properties and Microbial Community Structure

In the current study, the concentration of SOC and soil TN were significantly lower in the mixed-species plantation of *A. mangium* with native trees compared to the *A. mangium* monoculture in 2011 and in 2017 (Figure 5A,C). The results are different from many previous findings [14,16]. The differences might be related to the different vegetation biomasses between the mixed-species plantation and the monoculture in our study (Table 1). In general, the concentrations of SOC and soil TN were positively correlated with the vegetation biomass, because litter input mediated by vegetation biomass strongly contributed to the accumulation of SOC and soil TN [14,16,29]. As discussed above, the vegetation biomass was lower in the mixed-species plantations than in the *A. mangium* monocultures, leading to decreased input of C and N into the soil, thus, reducing SOC and soil TN accumulation.

The application of a PLFA analysis can provide a simple and fast procedure to estimate the microbial community composition [64,65]. The relative abundance of different microbial groups is identified, and thus, the microbial community composition can be roughly classified [64,65]. The relative abundance of all microbial groups (bacteria, fungi, actinomycetes, AMF, Gram-positive bacteria, and Gram-negative bacteria) in the *A. mangium* monoculture was comparable to the mixed-species plantation of *A. mangium*, leading to partial rejection of the second hypothesis relating to changes in microbial community structure under mixed-species plantation. Nonetheless, the ratio of fungi PLFA to bacteria PLFA in the mixed-species plantation is lower than that in the *A. mangium* monoculture (Figure 5J). This result may be related to the decreased soil TN concentration (Figure 5C), as the redundancy analysis shows that the relative abundance of fungi is positively correlated with soil TN concentration (Figure 6). This result was also proven in a previous study [66], which showed N additions significantly increased relative abundance of fungal PLFAs in an old-growth monsoon evergreen broadleaf forest in a subtropical region. Moreover, the difference of actinomycetes and AMF relative abundances between *A. mangium* monoculture and mixed-species plantation of *A. mangium* tended to increase with the increment of the planting year. Finally, we recommend that more long-term studies are required to deeply understand the dynamics of the soil microbial community in *A. mangium* monoculture and *A. mangium* mixed-species plantation.

## 5. Conclusions

Our findings highlight that *A. mangium* plantation is an appropriate option for degenerated land restoration with respect to soil rehabilitation, since it efficiently improves soil nutrient conditions within 33 years of planting. Our results showed that the improved soil N nutrient condition promoted N absorption by understory non-leguminous shrubs and herbs. Meanwhile, the increased soil AP concentration indicated that the planted N-fixing

species positively influenced soil P availability. However, *A. mangium* monoculture has disadvantages for new species regeneration and vegetation biomass increment, which could be compensated for by introducing native species into plantations. Interestingly, SOC and TN concentrations are lower in the mixed-species plantation of *A. mangium* with native trees than those in the *A. mangium* monoculture, while the soil microbial abundance is only slightly different between the mixed-species plantation and *A. mangium* monoculture after 19 years of the mixture. Finally, in order to further understand the effect of mixed-species plantation of *A. mangium* with native trees, many more longer term inventories are required.

**Supplementary Materials:** The following supporting information can be downloaded at: https://www.mdpi.com/article/10.3390/f14050968/s1, Table S1: List of the tree, shrub, and grass species in the plots of *A. mangium* monoculture and mixed-species plantation of *A. mangium*, Table S2: The sampling time, sampling plot, measured variable, and their objectives in this study, Table S3: Allometric equations for different species [67], Table S4: Two-way ANOVA analysis of planting year, plantation type, and their interaction on soil physiochemical properties, Table S5: Two-way ANOVA analysis of planting year, plantation type, and their interaction on relative abundance of soil microbial phospholipid fatty acids (PLFAs) (relative mol%).

**Author Contributions:** Conceptualization, W.S., X.C. and Y.L.; methodology, S.O., L.W., X.R., D.S. and S.L.; software, S.O., L.W., X.R., D.S. and S.L.; validation, W.S., X.C. and Y.L.; formal analysis, S.O. and L.W.; investigation, S.O., L.W., X.R., D.S. and S.L.; data curation, S.O., L.W., X.R., D.S. and S.L.; writing—original draft preparation, S.O.; writing—review and editing, A.G., A.K.B., L.T., V.V., W.S. and Q.Y.; visualization, S.O.; supervision, A.G. and W.S.; project administration, W.S., L.T. and S.O.; funding acquisition, W.S., L.T. and S.O. All authors have read and agreed to the published version of the manuscript.

**Funding:** This research was funded by the National Natural Science Foundation of China (31425005), the Natural Science Project of Guizhou University (2022-27 and 2022-31), and the Guizhou Provincial Science and Technology Projects (ZK[2022]YIBAN101, ZK[2023]YIBAN110).

**Data Availability Statement:** The data in this study are available from the authors upon request.

**Acknowledgments:** We would like to acknowledge the assistance from the staff of the Heshan National Field Observation and Research Station of Forest Ecosystem in China. We also gratefully thank Lixin Lyv and Abu Hanif for their valuable suggestions for improving the manuscript.

**Conflicts of Interest:** The authors declare no conflict of interest.

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
