# Peer review of "Mixed-Species Acacia Plantation Decreases Soil Organic Carbon and Total Nitrogen Concentrations but Favors Species Regeneration and Tree Growth over Monoculture: A Thirty-Three-Year Field Experiment in Southern China"

_forests, doi:10.3390/f14050968_

Round 1

Reviewer 1 Report

The paper is interesting and well written. The Authors have detailed precisely the objectives, the experimental design and the methods used. The results agree with the reported conclusions. The approach used is complete because it combines chemical soil and plant characterizations with biological/ecological aspects through the use of advanced analytical techniques such as the determination of PLFAs in order to evaluate the effect on soil biodiversity induced by association of the different tree species. Furthermore, the fact that it was possible to verify the effects over a rather long period of time is very appreciable. This certainly allows for greater robustness in the results obtained.

Author Response

 We thank the reviewer for the positive assessment. 

Reviewer 2 Report

The topic is not new, if considering monoculture plantations of leguminous tree species worldwide, but the manuscript meets the criteria of originality as a case study. The experiment was well done and properly documented, as well as the lab analyzes and statistics. Thus, the conclusions are clear and sufficiently documented in the Results section.

I don't have specific suggestions to the content, the manuscript is properly organised and written, so its reading is easy.

The only my suggestion is to resign of the term "laterite" (line 117), which is ambiguous in pedological terms. Of course, the material is highly weathered and acidic, but this does not specify the stage of pedogenic transformation. You correlated it with Oxisols, while FAO/ISRIC has correlated it - based on Chinese maps - with Acrisols (according to WRB classification) - you may check it easily in the website 

https://soilgrids.org/   (select Soil Classes)

For experts it makes a great difference. Thus, I suggest to resign of "laterite", while put the correlated soil units - Oxisols according to Soil Taxonomy and Acrisols - according to FAO/WRB classification.

Plase also check the line 63, an error is indicated there.

Author Response

Response letter

  1. The topic is not new, if considering monoculture plantations of leguminous tree species worldwide, but the manuscript meets the criteria of originality as a case study. The experiment was well done and properly documented, as well as the lab analyzes and statistics. Thus, the conclusions are clear and sufficiently documented in the Results section. I don't have specific suggestions to the content, the manuscript is properly organised and written, so its reading is easy.

Response: Thank you for the positive comments.

  1. The only my suggestion is to resign of the term "laterite" (line 117), which is ambiguous in pedological terms. Of course, the material is highly weathered and acidic, but this does not specify the stage of pedogenic transformation. You correlated it with Oxisols, while FAO/ISRIC has correlated it - based on Chinese maps - with Acrisols (according to WRB classification) - you may check it easily in the website 

https://soilgrids.org/   (select Soil Classes)

For experts it makes a great difference. Thus, I suggest to resign of "laterite", while put the correlated soil units - Oxisols according to Soil Taxonomy and Acrisols - according to FAO/WRB classification.

Response: Thank you for the good suggestion and useful link. We totally agree with the reviewer. We resigned the word ”laterite” and changed the description in the main text. The sentence was changed to “The soil is classified as acrisols based on WRB soil classification (WRB, 2015)”.

  1. Please also check the line 63, an error is indicated there.

Response: The word “of” was replaced by the word “with”.
